# Risk Perception of COVID−19 Community Transmission among the Spanish Population

**DOI:** 10.3390/ijerph17238967

**Published:** 2020-12-02

**Authors:** José Miguel Mansilla Domínguez, Isabel Font Jiménez, Angel Belzunegui Eraso, David Peña Otero, David Díaz Pérez, Ana María Recio Vivas

**Affiliations:** 1Department of Nursing, Faculty of Biomedical and Health Science, Universidad Europea de Madrid, 28670 Madrid, Spain; josemiguel.mansilla@universidadeuropea.es (J.M.M.D.); anamaria.recio@universidadeuropea.es (A.M.R.V.); 2Medical Anthropology Research Center, Faculty of Nursing, Rovira i Virgili University, 43002 Tarragona, Spain; angel.belzunegui@urv.cat; 3Hospital de Sierrallana, Advisor, Sub-Directorate of Primary Care, Cantabrian Health Service, 39011 Cantabria, Spain; david.penao@scsalud.es; 4Nursing Area, IiSGM Research Institutes Gregorio Marañón, 29007 Madrid, Spain; 5Nursing Area, IDIVAL Research Institutes Valdecilla, 39011 Cantabria, Spain; 6Respiratory Nursing Department at Sociedad Española de Neumología y Cirugía Torácica (SEPAR), 08029 Barcelona, Spain; ddiaperv@gobiernodecanarias.org; 7Pneumology and Thoracic Surgery Service of the Hospital Universitario Nuestra Señora de Candelaria, 38010 Tenerife, Spain; 8Coordinator of the Respiratory Nursing Department at SEPAR, 08029 Barcelona, Spain

**Keywords:** Covid19, SARS-CoV−2, community transmission, risk perception, public health

## Abstract

On 11 March 2020 the SARS-CoV−2 virus was officially declared a pandemic and measures were set up in various countries to avoid its spread among the population. This paper aims to analyse the perception of risk of COVID−19 infection in the Spanish population. A cross-sectional, descriptive observational study was conducted with a total of 16,372 Spanish participants. An online survey was used to gather data for 5 consecutive days over the compulsory lockdown period which was established after the state of emergency was declared. There is an association between socio-demographic variables and risk perception, and a very strong relationship between this perception and contact and direct experience with the virus in a family, social or professional setting. We also found that compared to working from home, working outside the home increased the perception of risk of infection and the perception of worsening health. Understanding the public perception of the risk of COVID−19 infection is fundamental for establishing effective prevention measures.

## 1. Introduction

According to the timeline established by the World Health Organisation (WHO), on 5 January the first report was published of an outbreak of pneumonia in China caused by a new virus. On 13 January, cases began to appear in areas outside China. On 30 January the new virus, dubbed SARS-CoV−2, was declared a public health emergency affecting 18 countries, and on 11 March it was officially declared a pandemic.

This novel coronavirus causes an illness called COVID−19, characterised as leading to serious pneumonia, although it is asymptomatic in 80% of cases [1]. Other typical symptoms include fever, dry cough, fatigue and aches [2,3], as well as gastrointestinal symptoms in 40% of cases, and other neurological and dermatological symptoms [4]. The patients considered to be the most vulnerable to developing symptoms are the elderly [5], due to the presence of immunosenescence [6], and people with existing conditions such as high blood pressure, cardiovascular diseases and diabetes [7].

It has been found to be highly contagious in the community, with transmission via respiratory droplets and direct contact [8]. Other studies discuss the possibility of aerosol transmission [9,10].

The first cases of COVID−19 in Spain occurred in February 2020 [11], in March, Europe became the epicentre of the pandemic [12]. France, Italy and Spain were the most affected countries with mortality rates of 15%, 13% and 11%, respectively. When analysing the data about mortality rate caused by COVID−19, it is important to consider that the different system of case registration among the European countries could have influenced the mortality rate differences.

Spain reached a peak on 3 April with 950 deaths per day (update No. 74). This situation posed a challenge for public health in these countries, which had to face the strong demand for health services, with a high number of hospital admissions and ICU admissions (26.5%) [13]. The recommendations supported personal hygiene measures, along with social distancing, as the most effective measures to stop the growth curve of cases [4,14,15]. In Spain, as had previously happened in Italy, mandatory confinement of the population was decreed on 14 March 2020 (Royal Decree 463/2020, 14 March) [16]. At that time, there were 7753 cases diagnosed in Spain, with a mortality rate of 3.7% of confirmed cases (Ministry of Health update No. 45).

The data on the number of infections, deaths and the saturation states of the hospitals was known by the population through different media, government information, social networks, television, radio, etc. Creating a social alarm that provoked fear among the population and a heightened risk perception [17,18] by associating the risk of infection with a real threat to their health [19].

At this time, it seemed interesting to inquire about monitoring preventive behaviours among the population and how these were associated or not with risk perception of SARS-CoV−2 contagion. This association seems interesting when observed in other pandemics as the relationship between risk perception and behaviour could offer information for the implementation of health policies, such as vaccination [20]. The authorities depend on monitoring the population for prevention measures to stop the spread of the virus [21,22]. Different factors studied in this pandemic increase risk perception, such as income level [23], vulnerability to the disease [24] and gender [25]. Other factors, such as lack of access to treatment or health services, are also associated with increased risk perception [26].

Risk perception belongs to the psychology category, which refers to an individual’s perception and understanding of the presence of objective danger to the individual or to the individual’s environment [27,28] Risk perception is an important factor that influences risk behaviours, such as whether or not to accept vaccination [29]. People with a lower risk perception tend to assume risky behaviours or reduce preventive behaviours, as shown by the Ding study, to the Chinese university population [21].

## 2. Materials and Methods

### 2.1. Study Design and Participant Selection

A cross-sectional, descriptive observational study was conducted with the participation of a total of 16,372 residents of Spain, 171 of whom were excluded for not meeting the criteria for inclusion (either they did not accept the informed consent statement, or they were under 18). In the end, the sample was 16,201 people (98.9% of the participants): 51.5% women and 48.5% men after weighting.

In order to get a large number of responses, we used non-probability consecutive sampling (snowball sampling). The initial bias of the sample was corrected by post-stratification, adjusting gross estimates according to gender and age, as proposed by Wang, Rothschild, Goel and Gelman [30]. Thus, in an appropriate statistical adjustment, non-random samples can generate accurate results, and refers to the importance of this type of survey for studying public opinion in different areas [30].

The adjustment method for the sample weighting was the iterative procedure with qualitative auxiliary information [31]. This procedure is used when there are qualitative auxiliary variables (gender and age groups) and maximum information is available on the univariate distributions of these auxiliary variables in the population and in the sample. Stratification is carried out by crossing these auxiliary variables, specifically from a double-entry contingency table. In the first iteration, the elevators corresponding to the gender variable are obtained and in the second iteration those corresponding to the age groups. Each iteration consists of two steps (as there are only two adjustment variables): (1) the distribution is adjusted with respect to the marginal population distribution of the first variable and (2) with this new weighted sampling distribution, the distribution is adjusted to the second characteristic. Finally, the assignment of elevators (Table 1) has enabled corrections to be introduced in other variables considered independent, such as educational level and marital status. The method also partially compensates for non-coverage [31].

### 2.2. Data Collection

An online questionnaire was used to gather data for 5 consecutive days over the compulsory lockdown period which was established after the state of emergency was declared in Spain. The questionnaire was distributed via social networks, scientific associations, and various healthcare-related institutions.

A specific questionnaire was prepared for this project using the tool “Google Forms”. The questionnaire comprised 59 items divided into 4 content areas (socio-demographic aspects, characteristics of the experience, health aspects, and risk perception) and included the project objectives, the requirements for participation and instructions for filling it in.

### 2.3. Variables

The dependent variable: to be studied was the perception of the risk of infection, a nominal variable with three categories (low, medium and high). The baseline variables which were considered as possible predictors of Perception of the risk of infection are as follows:

The Affect index is an ordinal variable (ranging in value from 0 = No affect to 1 = Maximum affect) constructed by standardising a set of diverse situations: a family member testing positive; a friend testing positive; somebody at work testing positive; death of a family member due to Covid−19).

Income was originally measured with an ordinal variable with 11 response categories, with extremes <€12,000 and >€300,000 euros. These categories were grouped into income quartiles: Q1: ≤€12,000; Q2: €12,001 to €22,500; Q3: €22,501 to €40,000; Q4: €40,001 to >€300,000.

The level of studies was classified into three levels: Primary studies (finished and unfinished); secondary studies (compulsory and post-compulsory) and higher studies (university).

The dichotomous variable work inside/outside the home was considered a possible predictor of risk perception since, remember, the survey was carried out in a period of confinement.

The coexistence variables (Who they live with) were also formulated dichotomously with response possibilities: Living alone and Living with others; if you live with a family member whose occupation is in healthcare (A family member works in healthcare, with Yes/No answers); Protective measures used (Protective measures, with Inadequate/Adequate categories); if Any family member has been affected (A member of the household is infected, with Yes/No answers).

Interviewees’ perception about whether their health had improved or worsened during confinement (*Health status during confinement*) was also considered a predictor variable, initially measured with a Lickert scale and later grouped into three categories (Good/Very good; Average; Poor/Very poor).

### 2.4. Analysis

Next, we conducted a preliminary analysis of the joint distribution of Perception of the risk of infection and the selection of potentially predictive variables (Table 2). The frequencies were calculated for each bivariate relationship and the corresponding relative frequencies were expressed as percentages. The hypothesis of independence between the variables was resolved with a Pearson chi-square test and its significance for a two-tailed distribution scenario. We also calculated the size of the effect of each relationship with Cohen’s *d*, as proposed by Rosenthal and DiMatteo [32], Lenhard and Lenhard [33] and Ferguson [34]. This analysis was used to create the final selection of variables which became part of the multinomial logistic regression analysis. All analyses were carried out with a confidence level of 95% and, therefore, a margin of error *p* = 0.05.

The multinomial logistic regression analysis was proposed in order to find predictors of the categories of Perception of the risk of infection, taking as the reference category Risk perception: Low. This type of analysis is especially suitable for modelling categorical and polytomous dependent variables, and is a multivariate extension of the binary logistic regression, admitting both continuous variables (covariates) and categorical variables (factors) as predictive variables [35].

The variables used in the model were gender, age, Affect index, place of work (working from home/outside the home), income (quartiles), health during lockdown (good/average/poor), protection measures (adequate/inadequate), whether a family member works in healthcare (yes/no), whether a member of the household is infected (yes/no) and where they get information on the disease (press/radio/TV; social networks/WhatsApp; official media/scientific documents).

We decided the model would not include some of the variables studied in previous analyses, such as education level, marital status, working/non-working population, and who they live with (living alone/with others); despite presenting statistical significance in the chi-square test, the coefficients of the size of the effect measured with Cohen’s *d* showed very small or negligible effects.

The parameters corresponding to the predictive variables were tested against the Wald test and its *p*-values, and the odds ratio (OR) of the coefficients and their confidence intervals were calculated. We also measured the goodness of the overall fit of the model with the chi-square test and the likelihood ratio test, and the percentage of correct classifications for each category. The goodness of fit was measured using the Nagelkerke pseudo R-squared parameter.

All the analyses were performed with the statistics program IBM SPSS v.21 (New York, NY, USA).

### 2.5. Ethical Aspects

The study was carried out in accordance with the principles of the Declaration of Helsinki and the laws and regulations in force in Europe and Spain and has been approved by the Research Ethics Committee University Europea (CIPI/20/138) and Research Ethics Committee with Medical Products of Cantabria (code: 2020.159).

Given the exceptional circumstances of the pandemic, and in line with the indications of the European Medicines Agency and the Ministry of Health, Consumer Affairs and Social Welfare, written informed consent was requested at the start of the online survey. It was imperative that the user accepted it in order to continue with the survey.

## 3. Results

First, the sample distribution is presented taking the variables considered in the analysis processes into account.

The Age variable produced an average age of 48.92 (95% CI: 46.68; 49.16) and Std. Error = 0.123. The average age of men was 48.79 (95% CI: 48.45; 49.14) and Std. Error = 0.175, and the average age of women was 49.04 (95% CI: 48.70; 49.38) and Std. Error = 0.174. The Affect index produced an average value of 0.3105 (95% CI: 0.3058; 0.3152), with Std. Error = 0.0023.

Table 3 shows the variables which were considered to be possible predictors of Perception of the risk of infection.

All the variables presented statistical significance, but for some the size of the effect was negligible. As proposed by Cohen (1988), effect sizes of <0.2 can be considered as No Effect. This is the case for gender (*d_Cohen_* = 0.1646), age (*d_Cohen_* = 0.1914), education level (*d_Cohen_* = 0.095), marital status (*d_Cohen_* = 0.084), impact on income (*d_Cohen_* = 0.1227 and who participants live with (*d_Cohen_* = 0.0407). Despite this, we decided to include gender in the model, as this variable contributes to including a gender perspective in the predictive model; and age, which we finally included not as a categorised variable but in its original form as a continuous variable.

Larger effect sizes corresponded to having a family member working in healthcare (*d_Cohen_* = 0.6051), degree of affect (*d_Cohen_* = 0.3276), the place of work (*d_Cohen_* = 0.4747), personal health perceived during lockdown (*d_Cohen_* = 0.3672), having an infected member of the household (*d_Cohen_* = 0.2516), and protection measures (*d_Cohen_* = 0.2478).

This first analysis could indicate an association between a higher perceived risk of infection and having a family member, friend or workmate test positive; working outside the home; perceiving one’s own health as poor during lockdown; someone in the household becoming infected; using adequate protective measures; and getting most of one’s information from official media and scientific documents.

Next, a prediction analysis of the categories of the dependent variable was performed through a multinomial logistic regression. In the analysis, a series of variables with statistical significance in the bivariate tests were selected, taking them as independent variables, and risk perception as the dependent variable. The multinomial model shows different OR values for the predictor variables, which offers an indication of the importance of these variables in determining risk perception.

The results of the model show an uneven predictive value: 82.2% for the category “Perception of the risk of infection: Low”, 15% for the category “Perception of the risk of infection: Medium”, and 49.0% for “Perception of the risk of infection: High”. The total percentage of correct prediction was 52.4% (Table 4), with a Kappa coefficient value of 0.223 (*p* < 0.001). The results show a significant model (−2log likelihood = 0.821; χ^2^= 2193.43; *p* < 0.001), with a moderate and acceptable goodness of fit Nagelkerke R^2^ = 0.204.

Table 5 and Table 6 show the regression coefficients of the predictive variables for the Medium and High categories, respectively. The comparison of the OR values (Exp(B)) of the two tables shows a higher predictive value in the independent variables considered in a high perception of risk. In fact, it is 4.439 times more probable that perception of the risk of infection will be high among those whose health has been poor or very poor during lockdown, compared to those who experienced their health as good or very good. It is also 4.251 times more probable that people with family members working in healthcare have a higher perception of the risk of infection than those without. Other relevant OR values correspond to people who work outside the home, compared to those working from home (OR = 3.057), those with someone infected at home compared to those without (OR = 2.167), and to a lesser degree, being a woman rather than a man (OR = 1.531), and getting most of their information from official media or scientific documents, compared to the conventional media, press, radio and TV (OR = 1.258). It is also important to mention the affect index variable, which presents an OR = 2.770, and which reflects that higher values of affect coincide with a higher perception of risk of infection. However, as the scanty association between age and perception of the risk of infection shown in Table 2 seems to indicate, the model does not produce a significant regression coefficient for age. All of these tendencies can be seen in the predictions for the category Perception of the risk of infection: Medium, but with lower intensity in all the variables.

To sum up, the multinomial model highlights better prediction in the category “Perception of the risk of infection: High”, compared to the middle category, which is more poorly defined, probably due to the tendency for those surveyed to place some items in middling positions when they have to do with assessments and/or perceptions of situations experienced during the COVID−19 lockdown.

## 4. Discussion

This study presents the risk perception of the Spanish public during the lockdown period coinciding with the maximum peak of infection of the pandemic. It shows an association of risk perception with socio-demographic variables and level of affect. Affect is understood as a variable of proximity to the disease, through direct contact or in the workplace, household or social circle.

A preliminary analysis shows that people who put inadequate prevention measures in place scored higher in low risk perception. Many studies refer to the important role of risk perception when establishing behaviour which will protect health in a pandemic [36,37,38,39] Specifically, the review by Bish and Michie [40] shows the association between perceived personal and family susceptibility to the development of a disease and the presence of preventive behaviour (hand washing, household hygiene, wearing masks, etc.) and avoidance behaviour (avoiding going to public places, restaurants, shops, etc.). Some of these papers place particular emphasis on the need to present concrete interventions and directives to the general public, or focused on specific groups, to help give them an outlook which strikes a balance between the risk they face and the risk they perceive [37]. For this reason, knowing what factors are associated with a greater perception of risk is fundamental when taking effective measures in a pandemic.

### 4.1. Influence of Socio-Demographic Variables on Risk Perception

The data indicate that women perceive more risk, in line with the study by Dryhurst et al. [22] which concludes that men perceive lower levels of risk than women. Women appear to be more susceptible to suffering from virus-borne diseases, and to present a greater incidence and severity of the disease [41]. Additionally, other studies conclude that more women are infected with SARS-CoV−2 via the main causes of infection: being a healthcare worker, visiting a medical centre, close contact with a person with an acute respiratory infection, direct contact with probable or confirmed cases of COVID−19 [42], or being an informal caregiver [43].

All of this seems to justify women having a greater perception of risk.

Bearing in mind that the elderly population are at greater risk of death [6,44], it is notable that this group, to a lesser degree, also has a high risk perception. These results are in line with the findings of Barber and Kim [25], in which they explain how elder adults report less concern and present less anxiety over health than younger adults; and they contradict other studies which determine that the elderly are one of the groups with a higher perception of threats [36,45].

### 4.2. Relationship of the Perception of the Risk of Infection with Contact with COVID-19 and Affect Level

A higher level of affect, understood as direct contact, correlates to an increased risk perception.

Direct contact with SARS-CoV−2 relates to a greater perception of risk; more specifically, having a family member working in healthcare, or having lived with somebody infected. Other studies support the results obtained, concluding that people who have had a direct personal experience with the virus, and people who have received more information about it, present a higher risk perception [22].

Meanwhile, it can be observed that workers who work outside the home have a high-risk perception. This is attested by studies such as Ruiz-Frutos et al. [46], which affirm that people working outside the home present higher levels of loss of sleep, unhappiness and depression. The authors associate this fact with the existence of a greater perception of risk of infection for the subjects or their families, compared to people working from home.

Many studies discuss the fact that healthcare workers’ heavy exposure to SARS-CoV−2 means they have an increased perception of the risk of infection [47,48,49], with nurses as a group showing a higher perception than doctors [50].

### 4.3. Correlation between the Perceived State of Health and Perceived Risk of Infection

We observe that people who perceive their own health as poor or very poor are the ones with the highest perception of risk [51,52]. However, they also present fear of going to hospital, demonstrated in a decrease in hospital admissions for such significant pathologies as myocardial infarction [53] and stroke during the lockdown period [54]. Additionally, multiple studies show that the presence of comorbidities increases the severity and risk of death of COVID−19 [55,56,57], a fact which may justify a sense of greater risk among the more vulnerable.

### 4.4. Limitations

The main limitation of this paper is that the sample was not randomised, limiting the general applicability of the results. Even so, the large size of the sample means we can regard it as fairly representative. The circumstances of the pandemic prevented the use of more accurate sampling methods, but nevertheless the results obtained are highly significant and could be the basis for future papers and work with randomised samples.

The fact that the sample is not comparable has likely led to bias in the response, highlighting the fact that more women than men and more people with university studies responded to it. This has partially been corrected using post-stratification methods, although the response bias is not completely eliminated. There continues to be a slight over-representation of people with higher education. This fact must be taken into account when interpreting the results.

This work has attempted to measure risk perception that each of the respondents considered they had and, therefore, the following question was formulated: What do you consider is your risk of infection? Where 1 is the minimum and 5 the maximum. It is a subjective measurement and, therefore, it is understood that it could vary depending on the knowledge and individual characteristics of each one of them. Therefore, even with no objective measurement of risk, this work focused on individuals’ subjective perception in an extreme situation of confinement. This variable reflected in the questionnaire thus enables the necessary data to carry out this study on the subjectivity of the impact of the disease to be obtained.

To carry out this study, our own survey was used, designed ad hoc and, therefore, with no specific sensitivity and specificity analysis. This survey does not contain any validated tool. Despite the design of the study does not allow to extrapolate the results to all the Spanish population, we consider that the population sample used makes that the data can be used as a guide to perform effective prevention strategies and developing new research projects.

Finally, it should also be noted that despite the statistical significance shown by some crossovers of variables, effect sizes <0.30 indicate that caution must be exercised when determining clear associations, so this represents an added limitation to the time to build strong relationships. In our analyses, this fact has been taken into account and clearly specified when low effect sizes were obtained.

## 5. Conclusions

Understanding the perception of the risk of COVID−19 infection is fundamental for establishing prevention strategies among the public.

An association can be observed between socio-demographic variables and risk perception, and a very strong relationship between this perception and contact and direct experience with the virus in a family, social or professional setting. The logistic model also finds that working outside the home increases the perception of risk of infection, compared to people working from home. Additionally, the perception of worsening personal health is strongly associated with greater perception of the risk of infection, as is living with a family member who works in healthcare.

### Applicability of the Results and Future Lines of Research

After carrying out this work, it is considered appropriate to develop new research projects aimed at knowing in detail what the behavioural and emotional responses of the population are, as well as their evolution during an epidemic or pandemic period. Specifically, one aspect that can be addressed in depth in future work is the fear of being stigmatised or discriminated against for having a positive test for COVID−19. A multidisciplinary perspective combines biological, psychological, and sociological aspects. Therefore, a multidisciplinary approach which also includes the psychological aspect, should be considered by the worldwide health policy makers [58]. It is also important to note that the groups that present a greater risk and fear perception are those that need the most support [59] and, therefore, require more complex investigations that place the focus of their objectives on it.

Dryhurst et al. identify the important role of risk perception in motivating health protection behaviours, especially during the COVID−19 pandemic [22]. They find that risk perception is positively and significantly correlated with an index of preventive health behaviours such as washing hands, wearing a mask and physical distancing and also point out that minimised as well as exaggerated risk perceptions can potentially undermine the adoption of behaviours to protect the [22]. Thus, higher risk perceptions lead to more protective behaviours, although taking effective measures can also reduce risk perception. Therefore, we highlight and suggest the importance of evaluating the accuracy of public risk perceptions and the correlation with taking positive protective behaviour measures. Among others, we should consider the vaccine acceptancy, being this one of the top 10 threats to global health in 2019 [60].

The providers of the different health systems in the different states must guarantee easy access to qualified information and support measures. Specifically, a transparent information policy that balances identifying risks, promoting appropriate behaviour and avoiding sensational reporting styles.

Policy-makers often conceptualise risk as the probability of contracting a disease multiplied by the magnitude of the consequences. However, our findings, which present evidence of how people perceived the risk of COVID−19 infection in Spain in the first wave, clearly illustrate that risk perceptions are consistently correlated with a number of experiential and sociocultural factors. More specifically, it is shown that risk perception among the population is higher in those with direct personal experience of the virus.

Therefore, health risk communication messages tend to be more effective when they include information on the effectiveness of measures aimed at protecting people from illness personally as well as socially.

Cori et al. indicate that knowledge implies the growth of collective awareness, increased self-efficacy and empowerment to contribute to political decision-making [61]. Mutual trust and dependence on local communication networks among peers could exponentially increase the possibilities to apply flexible measures, introducing concepts that promote health and well-being linked to healing, collaboration and solidarity. Therefore, the implementation of community interventions is recommended mainly in population groups in which risk perception is poorer.

Although this study is only observational and could be expanded with experimental studies, what does seem clear is that a better understanding of the knowledge that people have, as well as the social and cultural factors that accompany them, help to understand the risk perception of COVID−19 contagion in the world and its role in motivating preventive health behaviours. This information could help policymakers to design evidence-based risk communication strategies.

Therefore, future research is advised to consider expanding this research.

## Figures and Tables

**Table 1 ijerph-17-08967-t001:** Weights assigned to each substratum.

Age Groups	Men	Women	Total
Up to 30 years	2.8101	0.8942	1.3703
31 to 64 years	1.2379	0.5315	0.7424
More than 64 years	2.6100	3.8917	3.2073
Total	1.5635	0.7468	1.0000

**Table 2 ijerph-17-08967-t002:** Variables and distribution.

Perception of the Risk of Infection	*N*	%
Low	7481	46.2
Medium	5408	33.4
High	3312	20.4
**Age**		
Up to 30	2651	16.4
31−64	9700	59.9
65 and over	3850	23.8
**Gender**		
Female	8348	51.5
Male	7853	48.5
**Level of studies**		
Primary	1575	9.7
Secondary	5499	33.9
University graduates	9126	56.3
**Working population**		
No	4983	30.8
Yes		69.2
**Marital status**		
Single	4726	29.2
Separated/Divorced	1544	9.5
Married/Cohabiting	9376	57.9
Widowed	555	3.4
**Where they work**		
Working from home	6272	48.0
Working outside the home	6802	52.0
**Income: quartiles**		
First quartile	2519	16.5
Second quartile	5405	35.3
Third quartile	5729	37.4
Fourth quartile	1655	10.8
**Who they live with**		
Living alone	2047	12.6
Living with others		87.4
**Health status during confinement**		
Good/Very good		76.2
Average	3383	20.9
Poor/Very poor	464	2.9
**Protective measures**		
Adequate		71.9
Inadequate	4558	28.1
**You or relative work in healthcare** **A family member works in healthcare**		
No		76.6
Yes	3797	23.4
**A member of the household is infected**		
No		96.4
Yes	579	3.6
**How they get information**		
Press/Radio/TV		68.2
Social networks/WhatsApp	1521	9.4
Official media/Scientific documents	3630	22.4

Source: compiled by the authors.

**Table 3 ijerph-17-08967-t003:** Joint distribution between Perception of the risk of infection and different variables. *N* and percentages in brackets.

	Low	Medium	High	Pearson Chi-Square	*p* Value	Size Effect *d_Cohen_*
**Gender (*N* = 16,201**)						
Male	3933 (50.1)	2520 (32.1)	1400 (17.8)			
Female	3548 (42.5)	5408 (33.4)	3312 (20.4)	108.98	<0.001	0.165
**Age (*N* = 16,201)**						
Up to 30	1363 (51.4)	782 (29.5)	506 (19.1)			
31−64	4107 (42.3)	3450 (35.6)	2143 (22.1)			
65 and over	2011 (52.5)	1175 (30.5)	664 (17.2)	147.09	<0.001	0.191
**Education level (*N* = 16,201)**						
Primary	629 (39.9)	573 (36.4)	374 (23.7)			
Secondary	2580 (46.9)	1874 (34.1)	1045 (19.0)			
University graduates	4271 (46.8)	2962 (32.5)	1893 (20.7)	36.45	<0.001	0.095
**Affect (*N* = 13,813)**						
None	2556 (56.5)	1364 (30.2)	601 (13.3)			
Low	1746 (45.9)	1273 (33.4)	787 (20.7)			
Medium	1482 (42.5)	1191 (34.1)	818 (23.4)			
High	624 (41.2)	490 (32.3)	402 (26.5)			
Maximum	136 (30.6)	148 (33.3)	160 (36.0)	360.88	<0.001	0.328
**Marital status (*N* = 16,201)**						
Single	2260 (47.8)	1503 (31.8)	963 (20.4)			
Separated/Divorced	726 (47.0)	504 (32.6)	314 (20.3)			
Married/Cohabiting	4217 (45.0)	3256 (34.7)	1903 (20.3)			
Widowed	278 (50.1)	145 (26.1)	132 (23.8)	28.53	<0.001	0.084
**Working population (*N =* 16,194)**						
Non-working pop.	2736 (54.9)	1493 (30.0)	755 (15.1)			
Working pop.	4744 (42.3)	3913 (34.9)	2554 (22.8)	241.89	<0.001	0.246
**Place of work (*N* = 14,187)**						
At home	3341 (53.3)	2158 (34.4)	773 (12.3)			
Outside the home	2364 (34.8)	2354 (34.6)	2083 (30.6)	756.53	<0.001	0.475
**Economic situation (Impact on income) (*N* = 16,201)**						
Good	3731 (49.0)	2332 (30.6)	1550 (20.4)			
Average	2933 (43.8)	2368 (35.4)	1393 (20.8)			
Poor	817 (43.1)	708 (37.4)	369 (19.5)	60.70	<0.001	0.123
**Household income (*N* = 15,345)**						
1st quartile	1244 (49.4)	820 (32.6)	455 (18.1)			
2nd quartile	2338 (43.2)	1906 (35.3)	1162 (21.5)			
3rd quartile	2618 (45.7)	1888 (33.0)	1223 (21.3)			
4th quartile	791 (47.8)	494 (29.8)	370 (22.4)	41.68	<0.001	0.104
**Who they live with (*N* = 16,201)**						
Living alone	967 (47.2)	634 (31.0)	446 (21.8)			
Living with other(s)	6513 (46.0)	4774 (33.7)	2867 (20.3)	6.70	0.035	0.041
**Health (*N* = 16,190)**						
Good/Very good	6192 (50.2)	4027 (32.6)	2125 (17.2)			
Average	1169 (34.6)	1248 (36.9)	966 (28.6)			
Poor/Very poor	117 (25.2)	132 (28.4)	215 (46.3)	527.86	<0.001	0.367
**Protective measures used (*N* = 16,201)**						
Adequate	5012 (43.0)	3927 (33.7)	2704 (23.2)			
Inadequate	2469 (54.2)	1481 (32.5)	609 (13.4)	245.04	<0.001	0.248
**You or relative work in healthcare** **Family member works in healthcare (*N* = 16,201)**						
No	6375 (51.4)	4274 (34.5)	1755 (14.1)			
Yes	1105 (29.1)	1134 (29.9)	1558 (41.0)	1358.72	<0.001	0.605
**Household member infected (*N* = 16,201)**						
No	7334 (46.9)	5241 (33.6)	3046 (19.5)			
Yes	146 (25.3)	166 (28.7)	266 (46.0)	252.43	<0.001	0.252
**Where they get information (*N* = 16,201)**						
Press/Radio/TV	5263 (47.6)	3788 (34.3)	1998 (18.1)			
Social networks/WhatsApp	741 (48.7)	464 (30.5)	316 (20.8)			
Official media/Scientific documents	1477 (40.7)	1156 (31.8)	998 (27.5)	236.99	<0.001	0.244

Source: compiled by the authors.

**Table 4 ijerph-17-08967-t004:** Table classifying the prediction of the categories of Perception of the risk of infection in the multinomial logistic regression model.

Classification	Predicted			
Observed	Low	Medium	High	Percent Correct
Low	4090	484	401	82.2%
Medium	2558	562	618	15.0%
High	897	326	1176	49.0%
Overall Percentage	67.9%	12.3%	19.8%	52.4%

Source: compiled by the authors.

**Table 5 ijerph-17-08967-t005:** Parameters of the multinomial logistic regression analysis for the category “Perception of the risk of infection: Medium”. The reference category is: Low.

Risk of Infection: Medium	B	Std. Error	Wald	Sig.	OR	95% CI for OR	
						Lower Bound	Upper Bound
Intercept	−0.938	0.099	89,821	0.000			
Age	0.001	0.002	0.388	0.534	1001	0.998	1004
Affect (index)	0.502	0.08	38,983	0.000	1652	1411	1935
Gender (female)	0.291	0.045	42,401	0.000	1337	1225	1459
Working outside the home	0.457	0.046	970.21	0.000	1579	1442	1729
Health: poor/very poor	0.491	0.164	8997	0.003	1634	1185	2252
Health: average	0.455	0.059	590.499	0.000	1577	1405	1770
Protective measures: adequate	−0.173	0.048	12,778	0.000	0.841	0.765	0.925
Family member works in healthcare	0.347	0.059	35,115	0.000	1415	1261	1586
A member of the household is infected	0.244	0.145	2855	0.091	1277	0.962	1696
Information: official media	0.030	0.056	0.030	0.584	1031	0.925	1149
Information: social networks	−0.236	0.078	9134	0.003	0.790	0.678	0.921

Source: compiled by the authors.

**Table 6 ijerph-17-08967-t006:** Parameters of the multinomial logistic regression analysis for the category “Perception of the risk of infection: High”. The reference category is: Low.

Risk of Infection: High	B	Std. Error	Wald	Sig.	OR	95% CI for OR	
						Lower Bound	Upper Bound
Intercept	−2699	0.132	415,322	0.000			
Age	0.004	0.002	3261	0.071	1004	1000	1008
Affect (index)	1019	0.101	102,291	0.000	2770	2274	3375
Gender (female)	0.426	0.056	57,432	0.000	1531	1372	10.71
Working outside the home	1117	0.061	335,398	0.000	3057	2712	3445
Health: poor/very poor	10.49	0.163	83,237	0.000	4439	3223	6115
Health: average	0.788	0.069	130,468	0.000	2210	1922	2519
Protective measures: adequate	−0.647	0.067	94,344	0.000	0.524	0.46	0.597
Family member works in healthcare	1447	0.062	539,488	0.000	4251	3762	4803
A member of the household is infected	0.773	0.142	29,649	0.000	2167	10.64	2862
Information: official media	0.229	0.066	12,163	0.000	1258	1106	1431
Information: social networks	−0.046	0.095	0240	0.624	0.955	0.793	1149

Source: compiled by the authors.

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
