# Peer review of "Risk Perception of COVID−19 Community Transmission among the Spanish Population"

_ijerph, 2020, doi:10.3390/ijerph17238967_

Round 1

Reviewer 1 Report

The manuscript "Risk perception of SARS-CoV-2 community transmission among the Spanish population" summarizes a cross-sectional surveys on risk perceptions among Spanish citizens during enforced social distancing control measures. It is important to monitor risk perceptions and risk behaviours during emergency response. However the contribution should be more carefully presented and clearly structured. There are also important methodological flaws which are not sufficiently addressed. I therefore suggest a major revision of the manuscript,

1. The Introduction is quite chaotic. It gives a lot of facts on the development of the epidemic and the context of the investigation, but each piece of information is fragmented and the entire section lacks structure that helps introduce the study aim. I think a good introduction could be structured to answer three questions: 1) what burden Covid-19 places on the society (not necessarily only Spanish), 2) what happened in Spain but keeping the chronology and avoiding interpretations and conclusions, 3) introducing the concept of risk perception and why it is important to investigate it. Ideally, the Introduction should be kept concise, but could be longer than it is now.

2. The authors correctly emphasize that convenience sample was the main limitation of this investigation. Especially it could influence their results of risk perceptions which could be very different in specific populations, confined geographically and their snowball sampling could actually lead to recruit only one type of respondents which share specific beliefs and behaviours and are happy to respond to such survey, because they can be particularly interested in the topic. Post-stratification could be a solution if an effort would be documented to include residents of all types of social environments in the entire country. Not only there is no such documentation, but the adjustment method is not explained in detail. I recommend putting more attention to address this key limitation which could undermine the results of this investigation.

3. The Methods section includes results, especially in the section «Variables». This is making the manuscript more difficult to follow, because the reader expects to understand the methodological choices, the construction of measurement tools, etc., in this section. This is a good idea to list the variables in a table. However, I strongly recommend to use the methods to focus on the construction of the variables, ie. categories used, groupings of combinations of variables or their categories.

4. In the Results, I recommend starting by a thorough description of the population, which could include some of the information included in the Methods (for example, in Table 1).

5. The multivariable analysis could be better introduced and presented, to allow the reader to follow the authors intentions. The coefficients are not intuitively understood. Therefore, I suggest simplifying the tables 4 and 5 to include clearly labelled odds ratios with confidence intervals. The full outputs could be included in a supplemental file.

6. The authors claim that this investigation could provide evidence informing the public interventions. I recommend to emphasize more clearly the public health implications of their results and add their recommendations on how their results should be applied to guide public health intervention in a short and long-term. This section should directly follow the conclusions and should be directly link to them.

Author Response

Thank you for agreeing to review this manuscript, the contributions you have made enable this work to be improved. We respond to your contribution in the attached document. Changes to the text are shown in yellow.

Reviewer 2 Report

Re: Risk perception of SARS-CoV-2 community transmission among the Spanish population

Thank you very much for offering me the opportunity of reviewing this paper. The paper used data from an online questionnaire to examine the perception of risk of SARS-COV-2 infection in the Spanish population. the research question is sound and important in the research field. However, there are a few major issues of this manuscript, which needs to be solved before publication. My comments are below:

Major concern:

  1. The authors could consider use the term “COVID-19” (named by WHO) instead of “SARS-Cov-2” to keep consistency in the literature and avoid confusion.
  2. There are some studies which have been conducted on mental health in COVID-19 pandemic (OR SARS-Cov-2 pandemic mentioned by the authors in the manuscript). They should be reviewed and the findings should be summarized in the introduction section. The literature review is not comprehensive in the current version of the manuscript.
  3. The conceptualization of risk perception can be strengthened in the introduction section.
  4. Instead of using the term “various studies”, the authors are recommended to provide a more detailed summary of the literature, and summarize the research gaps in the literature.
  5. The context of Spain during COVID-19 pandemic should be further discussed in the introduction section. Such information is important for the audience to understand the risk perceptions of COVID-19.
  6. The specific procedures of the sampling strategies should be further discussed. More information should be provided, such as response rate, and missingness. Online survey has both merits and limitations, which deserve further discussions in the sampling section and limitation section (e.g., information accuracy).
  7. The differences between SD and SE should be double checked.
  8. Socio-demographic characteristics of respondents are generally listed in the Table 1 in the result section. Measurement section should focus on the description of the measures.
  9. Research references should be provided in the data analysis section.
  10. There could be an issue of observation dependence if people from same families (e.g., couples) participated in the online survey. Sensitivity analysis should be conducted.
  11. Exp (B) in table 4 and table 5 should be changed to “OR”
  12. Policy and intervention implications of the findings should be further elaborated.
  13. The limitation section is too brief.

Author Response

(The authors gave the same response as above.)

Reviewer 3 Report

Dear Authors,

thanks for proposing this contribution regarding SARS-CoV-2 pandemic.

  1. Please provide me the Google Form questionnaire used within the research study; does it contain some validated tools?
  2. Did you ask the workers if they worked in healthcare or did you just investigate any family members in the healthcare sector? have you discarded them?
  3. I think you should update the introduction section with the latest data regarding the second flow and consider it into discussion and conclusion sections.
  4. You did not investigate the fear of being stigmatized or discriminated against if they test positive or a family member / close cohabitant tests positive for SARS-CoV-2. This is a limitation that you may consider 
  5. What if the risk perception is inappropriate?

Please update these gaps referring to the following references:

  • Sauer, K.S.; Jungmann, S.M.; Witthöft, M. Emotional and Behavioral Consequences of the COVID-19 Pandemic: The Role of Health Anxiety, Intolerance of Uncertainty, and Distress (In)Tolerance. Int. J. Environ. Res. Public Health 2020, 17, 7241
  • Irigoyen-Camacho, M.E.; Velazquez-Alva, M.C.; Zepeda-Zepeda, M.A.; Cabrer-Rosales, M.F.; Lazarevich, I.; Castaño-Seiquer, A. Effect of Income Level and Perception of Susceptibility and Severity of COVID-19 on Stay-at-Home Preventive Behavior in a Group of Older Adults in Mexico City. Int. J. Environ. Res. Public Health 2020, 17, 7418
  • Baldassarre, A.; Giorgi, G.; Alessio, F.; Lulli, L.G.; Arcangeli, G.; Mucci, N. Stigma and Discrimination (SAD) at the Time of the SARS-CoV-2 Pandemic. Int. J. Environ. Res. Public Health 2020, 17, 6341
  • Ding Y, Du X, Li Q, Zhang M, Zhang Q, Tan X, et al. (2020) Risk perception of coronavirus disease 2019 (COVID-19) and its related factors among college students in China during quarantine. PLoS ONE 15(8): e0237626
  • Sarah Dryhurst, Claudia R. Schneider, John Kerr, Alexandra L. J. Freeman, Gabriel Recchia, Anne Marthe van der Bles, David Spiegelhalter & Sander van der Linden (2020) Risk perceptions of COVID-19 around the world, Journal of Risk Research, DOI: 10.1080/13669877.2020.1758193
  • Wise, T., et al. (2020) Changes in risk perception and self-reported protective behaviour during the first week of the COVID-19 pandemic in the United States. Royal Society Open Science. doi.org/10.1098/rsos.200742

Author Response

(The authors gave the same response as above.)

Reviewer 4 Report

Identifying the risk perception in a community/ country is essential as much as educating prevention measures. 

However, I would recommend some revisions.  

  1. In the title, SARS-CoV-2 is the causing virus of COVID-19. Do people fear the virus or infection? 
  2. In “Materials and Methods,” the third and fourth paragraphs need to be combined and categorized by ‘Data collection.’ 
  3. You may avoid redundant statements by deleting Line 91~93.
  4. Please reconsider to use options such as widowed/separated/divorced to ask their marital status. (in terms of ethical issues) 
  5. How about combining Table 1 and Table 2 ?
  6. In tables, please recheck how to report p-value and size effect. 

(p-value, 0,000 →<.001; size effect, 0.1646→0.16)  

    7. In “Limitations,” Line 265~267 needs to move to “Materials and Methods.” 

    8. Please add your suggestions for further studies and policy development. 

Overall, this manuscript needs a more concise statement and clarity.   

Author Response

(The authors gave the same response as above.)

Round 2

Reviewer 1 Report

The authors have addressed the suggestions, therefore I recommend to publish the contribution.

Author Response

Thank you for accepting our study.

Reviewer 2 Report

I have no further comments.

Author Response

Thank you for accepting our study.

Reviewer 3 Report

Dear Authors,

still some points to deepen:

  1. line 50 - please refer to the different system of case registration in different European countries, which could have influenced the mortality rate and would make it difficult to make comparisons, which will be possible only at the end of the epidemic.
  2. line 325 - please specify your ad hoc questionnaire does not contain any validated tools
  3. line 326 - how do you assert that "this work can be taken as a reference in terms of risk perception among the Spanish population" ? I believe it is an observational study conducted without the basis to be able to assert this.
  4. line 326 - work just doesn't sound good... express yourself slavishly to the scientific context, maybe "study" or "research"?
  5. please add 0 in formulas (p<0.05 or <0.30)
  6. line 350 - please refer to https://doi.org/10.3390/ijerph17176341 and explain why is crucial an multidisciplinary approach and preventive information by worldwide health policy makers to face stigma and discrimination as happened for previous health emergencies
  7. I would like you to also hint at the future prospect of the vaccine and what this might imply in terms of hesitation on the part of the general population, as anticipated in 2019 by the WHO (please also refer to https://www.who.int/news-room/spotlight/ten-threats-to-global-health-in-2019 ) 

not having used a validated questionnaire or a tailored one containing references to other validated questionnaires unfortunately lowers the outcome level, which is still sufficient.

I request minor revisions but I consider the criticality just expressed at point 3 as major.

Author Response

Thank you for agreeing to review this manuscript, the contributions you have made enable this work to be improved. Changes to the text are shown in yellow.

Reviewer 4 Report

I appreciate the authors for their efforts on revisions.

Author Response

Thank you for accepting our study.